# On-Site Health Monitoring of Composite Bolted Joint Using Built-In Distributed Eddy Current Sensor Network

**DOI:** 10.3390/ma12172785

**Published:** 2019-08-29

**Authors:** Qijian Liu, Hu Sun, Tao Wang, Xinlin Qing

**Affiliations:** School of Aerospace Engineering, Xiamen University, Xiamen 361005, China

**Keywords:** composite, bolted joint, structural health monitoring, built-in, eddy current sensor network

## Abstract

There is an urgent need to monitor the structural state of composite bolted joints while still remaining in service; however, there are many difficulties in analyzing their strength and failure modes. In this paper, a built-in distributed eddy current (EC) sensor network based on EC array sensing film is developed to monitor the hole-edge damages of composite bolted joints. The EC array sensing film is bonded onto the bolt and consists of one exciting coil and four separate sensing coils. Experiments are conducted on unidirectional composite specimens to validate the ability of the EC array sensing film to quantitatively track the damage that occurs at the hole edge and to investigate the performances of the EC array sensing films with different configurations of the exciting coil. Experimental results show that the induced voltage of sensing coil changes only if the damage appears on the laminate structure where that particular sensing coil is located, whereas the induced voltages of the other sensing coils on other laminate plates remain unchanged. Numerical simulation based on the finite element method is also carried out to investigate and explain the phenomena observed in the experiments and to analyze the distribution of the EC around the bolt hole. Both experimental and numerical simulation results demonstrate that the developed EC array sensing film can effectively identify not only whether there is damage at the hole edge but also the damage location within the thickness and quantitative size.

## 1. Introduction

Fiber-reinforced composite materials are gaining more and more importance in aerospace, automotive, shipbuilding, and other industries due to their high strength-to-weight and stiffness-to-weight ratios. By utilizing these advanced composite materials, large aircrafts are able to improve its efficiency while reducing its weight and operational costs [1]. As one of the key components of large composite structures, joints are widely used to ensure the integrity of composite structures. The mechanically fastened joint is widely used for connections within the composite structure due to its exceptionally strong bearing capacity, high reliability, and ease of disassembly and maintenance. The structural integrity of bolted joints can generate a significant impact on the stability and safety. Furthermore, the knowledge of the failure load and the response of bolted joints is critical for the design of the structures [2,3,4]. Nevertheless, it is very difficult to solve the strength and failure modes of composite bolted joints due to multiple factors. Due to the structural importance of joints, material degradation and other types of damages in the joint is a major concern. Consequently, timely and accurate damage detection along with active characterization and monitoring of delamination is needed. Damages in composite structures can be much more difficult to inspect and evaluate compared to metallic structures. Conventional nondestructive inspection techniques utilizing a variety of methods such as ultrasonic C-scan, eddy current, thermography, and shearography are difficult to perform during operation due to the inaccessibility of the joint. Therefore, there is an urgent need to develop a method to monitor the structural state of the composite bolted joints while staying in service.

By using a built-in sensor network integrated with the structure, structural health monitoring (SHM) can supply crucial information corresponding to the current condition and damage state of the structure [5,6,7,8,9,10,11,12,13]. SHM is showing great promise of being embraced by the industries as a capable method of monitoring the structural condition throughout its lifetime without requiring disassembly. An overview of structural health monitoring of composite joints can be found in the literature [14]. Several SHM technologies have been developed for damage inspection of bolted joints. Jalalpour et al. [15] made a combination of ultrasonic signals with fuzzy pattern recognition to monitor the integrity of 90° bolted joints. Rakow and Chang [16] developed a flexible eddy current (EC) sensing film bonded on the bolt to detect the existence of bolt hole-edge crack and its propagation. Sun and Qing et al [17,18] further developed the EC array sensing film bonded on the bolt to quantitatively monitor the crack growth along the radial direction and axial direction of the bolt hole. Some other methods, including acoustic emission [19], acoustic-ultrasonics [20], electromechanical impedance [21], comparative vacuum monitoring (CVM) [22], and intelligent coating monitoring (ICM) [23], can also be used to inspect the damage of bolted joints. However, these researches with flexible EC sensing film are focused on the metallic joints because of their good conductivity. 

The concept of a multi-field coupled sensing network based on piezoelectric sensors, EC sensors, and the Rogowski coils was developed to allow the structural state to be actively monitored while the structure remains in service [24]. However, it is difficult to construct a multi-field coupled sensing network for composite bolted joints due to some challenges and issues, including how to design the EC sensors for composite joints and how to integrate a variety of sensors together. In this study, a built-in distributed EC sensor network based on EC array sensing film is developed to quantitatively monitor the damages around the bolt hole of composite joints. The anisotropic conductivity of carbon fiber reinforced plastic (CFRP) is analyzed and employed in the numerical simulation. The principles and configuration designs of EC sensing film are first described. Then a series of experiments are conducted on unidirectional composite specimens to prove the feasibility of EC array sensing film to track the hole-edge damage and to investigate the performances of EC array sensing films in the condition of different exciting coil configurations. Numerical simulation is finally carried out to investigate and explain the phenomena observed in the experiments.

## 2. Principles and Configuration Design of Eddy Current Sensing Film

### 2.1. Anisotropic Conductivity Property of CFRP

The structure of CFRP consists of carbon fibers and resin matrix. The CFRP offers excellent electrical conductivity along fiber direction. Additionally, considerable transverse electrical conductivity is also observed in carbon fiber composites since there are several contact points between fibers during the high-temperature curing process of prepregs. In other words, the composite materials are electrically conductive in all directions. Different orientations and cross-connections of carbon fibers will influence the mechanical and electrical properties of CFRP. Furthermore, the volume fraction of carbon fiber also affects the conductivity of CFRP. In general, the volume fraction of carbon fiber in CFRP is about 60%–70%. 

Figure 1a shows a schematic diagram of contact points between carbon fibers, and Figure 1b represents the current paths between carbon fibers in CFRP materials. The contact between the adjacent fibers constructs a closed conductive loop in CFRP and the current can flow along the path. However, the conductivity of composite material will change when damage appears. For example, fiber fracture or structure failure will cause resistance to increment due to the breakage of fibers or connection points between fibers. 

The electrical conductivity in CFRP can be expressed in three directions (longitudinal, transversal, and through-thickness). The value of conductivity in three directions are presented as $\sigma_{L}$, $\sigma_{T}$, and $\sigma_{\mathrm{cp}}$. The longitudinal conductivity, $\sigma_{L}$, is between $5\times 10^{3}$ and $5\times 10^{4}$ S/m along the fiber. The transversal conductivity, $\sigma_{T}$, varies from 10 to 100 S/m [25]. The plies of CFRP laminates are contacted during the manufacturing process which provides a through-thickness conductivity between the adjacent plies. Nevertheless, the conductivity between each ply of CFRP, $\sigma_{\mathrm{cp}}$, is much smaller than the transversal direction. The expression of anisotropic conductivity of CFRP can be represented as a conductivity tensor relied on the rotated coordinate system (Figure 2). The generalized definition of conductivity tensor is calculated in a matrix as shown in Equation (1), and θ is a parameter relative to the fiber orientation.
(1)$$\overset{=}{\sigma }=\left[\begin{array}{ccc} \sigma_{L}\cos^{2}\left(\theta \right)+\sigma_{T}\sin^{2}\left(\theta \right) & \frac{\sigma_{L}-\sigma_{T}}{2}\sin\left(2\theta \right) & 0\\ \frac{\sigma_{L}-\sigma_{T}}{2}\sin\left(2\theta \right) & \sigma_{L}\sin^{2}\left(\theta \right)+\sigma_{T}\cos^{2}\left(\theta \right) & 0\\ 0 & 0 & \sigma_{\mathrm{cp}} \end{array}\right]$$

### 2.2. Principles of Monitoring

The basic principles of monitoring using the EC sensor are shown in Figure 3. The EC sensor is manufactured on a flexible substrate with printed flexible circuit technology and bonded on the bolt of composite joint. To achieve the quantification of monitoring a bolt hole-edge damage of both radial and axial directions, the EC sensing film is formed by an exciting coil and several sensing coils [17]. According to Faraday’s Law, when an alternating electric current is applied to the exciting coil, a primary alternating magnetic field will be produced and surround the exciting coil, and then an EC will be generated in the conductive composite material. The EC will generate a secondary alternating magnetic field, restraining the original magnetic field. When damage occurs at the bolt hole-edge, the generated EC will be disturbed due to the change of the original flow path and consequently, will also affect the secondary magnetic field. By analyzing the signals captured from the sensing coil, hole-edge damages in the composite laminates can be identified.

## 3. Experimental Investigation

A series of experiments were designed to prove the capability of the EC array sensing film to quantitatively track the damage around hole edge, and the performances of the EC array sensing films were investigated with different configurations of the exciting coil. 

Each EC array sensing film used in the experiments has one exciting coil covering the entire region of the inner wall and four separate sensing coils laying along the axial direction of the bolt hole. As shown in Figure 4, each sensing coil is constructed with two sections of traces, which are separated by the dielectric film and connected through welding at the weld hole near the center of the traces. The exciting coil has the same structure as sensing coils. Two different configurations of the exciting coil were investigated in the experiments. Both of the two configurations are similar to the sensing coil, whereas all of the exciting traces are in the series connection as one coil. The main difference between the two configurations of exciting coil lies in the direction of current path on the boundary of two adjacent exciting sub-coils. In one configuration, the direction of current at the boundary of two adjacent sub-coils is the same, while the opposite current direction is set in the other configuration. Figure 4 shows the plot of exciting coil with the exciting current in the boundary trace of one exciting sub-coil being in the same direction as that in the adjacent boundary trace of another exciting sub-coil, which can be achieved by applying the current from external circuits. 

### 3.1. Experimental Set-Up 

The EC array sensing films with different configurations of exciting coil described above were mounted on the bolts, respectively. Both the width of each trace and the distance between all of the traces are 0.25 mm. The whole structure of the EC sensing film was comprised of sensing coils with 20 traces and an exciting coil with 80 traces. The parameter for the height and the length of the EC array sensing film is 20 mm and 43.96 mm, respectively. The diameter of the bolt hole is 14 mm and the thickness of the CFRP specimens is 20 mm. 

Figure 5 shows the detailed information regarding the experimental setup. A sinusoidal signal with a frequency of 8 MHz and output voltage of 1.2V was generated by a signal generator, Rigol DG 3061A (RIGOL Technologies USA, INC., Beaverton, OR, USA), which was amplified by an AG1020 RF amplifier (T&C Power Conversion, Inc., Rochester, MN, USA) and then inputted to the exciting coil. The induced voltage of sensing coils was measured by a lock-in amplifier (SYSU SCIENTIFIC INSTRUMENTS, Guangzhou, China), OE2041. The values of induced voltage from different sensing coils were obtained by the switch used to connect each coil.

The experiments on unidirectional CFRP specimens made of carbon fiber T300 prepreg were conducted. The length, width, and thickness of unidirectional CFRP specimens are 100 mm, 100 mm, and 20 mm, respectively. In order to make artificial defects accurately, the joint was designed to consist of four laminate plates, each having a 5 mm thickness. The stacking sequence of each laminate plate is [0/45/90/−45]_3s_.

During the experiment, artificial cracks (notches) with different lengths were introduced at the hole edges of unidirectional CFRP specimens, as shown in Figure 6. The cracks were increased in the radial direction of the bolt hole by 1 mm each step, or in the axial direction by 5 mm each step. The variation of induced voltages related to the cracks propagating along the axial direction and radial direction were investigated.

### 3.2. Experimental Results

Figure 7 presents the changes of induced voltages of the sensing coils, caused by the damage at the hole edge in the top laminate plate when increased from 0 mm to 4 mm along the radial direction of the bolt hole of the joint. From the results, the induced voltage of sensing coil 1 at the top laminate plate raised rapidly when the crack occurred at the top laminate plate. In the meantime, the induced voltages from the other three sensing coils located at the other laminate plates without cracks remained unchanged. The results indicate that the values of the induced voltages are higher when the exciting current in the boundary trace of one sub-coil is in the same direction as that in the adjacent boundary trace of another sub-coil compared to when the adjacent boundary trace currents move in opposite directions. This is because the intensity of current is increased when the current in the boundary traces of adjacent coils is in the same direction.

The changes of induced voltages from the sensing coils caused by the artificial crack at the hole edge in the second laminate plate when increased from 0 mm to 4 mm along the radial direction of bolt hole are depicted in Figure 8. Similar to the results shown in Figure 7, the induced voltage from sensing coil 2 at the second laminate plate raised rapidly when the crack occurred in the second laminate plate, while the induced voltages from the other three sensing coils located at the other laminate plates remained unchanged. 

Furthermore, Figure 7 and Figure 8 illustrate that the change in the induced voltage of the sensing coil at the damaged area continuously increases when the crack propagates to about 3 mm. Passed that, the induced voltage of the sensing coil remains unchanged, even as the crack propagates further. This is because the EC mainly distributes near the surface of the bolt hole, and the penetration depth of the EC is limited.

Figure 9 presents the changes of induced voltages corresponding to the crack of 2 mm in the radial direction propagating in the axial direction along the bolt hole from 0 mm to 20 mm. The crack was increased from 0 mm to 20 mm with 5 mm per step. It is obvious that the value of induced voltage increases only in the sensing coil at the location where the damage appears, while the value of the remaining sensing coils only changes slightly or are unvaried. As mentioned previously, the response of the induced voltage is more sensitive for the configuration of when the exciting current in the boundary trace of one exciting sub-coil is in the same direction as that in the adjacent boundary trace of another exciting sub-coil than when the adjacent boundary trace currents point in opposite directions. The damage destroys the path of current flow in CFRP, and the intensity of the secondary magnetic field generated by EC decreases. Consequently, there is an increment of total magnetic field based on subtraction of the original magnetic field and the secondary magnetic field. This is the reason why the induced voltage increases when the crack occurs in both axial and radial directions.

## 4. Numerical Simulation

Preliminary numerical simulation was conducted to verify and explain the phenomena observed in the experiments. Two types of artificial cracks around bolt hole, one propagating along the axial direction and the other propagating along the radial direction are introduced in the CFRP specimen to validate the change of the induced voltage from the EC array sensing coils.

### 4.1. Numerical Model

A three-dimensional simulation model based on the finite element method (FEM) has been developed to determine the changes of induced voltages of EC sensing coils resulting from the cracks propagating around the hole-edge in the composite bolted joints. The configuration of the FEM model is shown in Figure 10. The dimensions of CFRP composite specimen are 30 mm × 30 mm × 20 mm with one-quarter of a hole having a diameter of 7 mm. In order to facilitate calculation and save time, the model used only contains one-quarter of a circular hole because the distribution of EC around the hole is symmetrical. The layup of CFRP composite is [0/45/90/−45]_12s_ All adjacent plies in the model are assumed to be in contact with each other.

As shown in Figure 10, the inner traces are four sensing coils and the outer traces are the exciting coil with four exciting sub-coils. The flow path of the current for the nearby traces of the exciting coil at the boundary of two adjacent coils is set in the same direction in the simulation. Alternating current with a frequency of 8 MHz and 2 V is applied to the exciting coil. The mesh of the FEM model is generated with tetrahedral elements. The conductivity in the fiber direction is 29300 S/m, the conductivity perpendicular to fiber is 77.8 S/m, and the conductivity in the thickness direction is 0.000794 S/m. The crack was set to increase 2.5 mm each step in the axial direction, and 0.5 mm each step in the radial direction.

### 4.2. Simulation Results

The EC distributions on the first four plies [0/45/90/−45] from the top of CFRP composite laminates with and without a crack is shown in Figure 11 and Figure 12, respectively. Under the simulation condition, the distribution of the EC for each ply is not the same because of the effect of the fiber orientation. For the plies with fiber direction of 0° and 90°, the distribution of EC is concentrated in the position of the opposite sides of the hole edge. The EC flows in the opposite direction when located on the ply with fiber orientation in 45°. The EC repartition of ply with fiber direction of −45° is mainly concentrated at the central area of the hole edge. When the artificial crack set at the central area of the hole edge, the EC distribution of ply with fiber direction of −45° changed dramatically due to the current density decreasing. This indicates that the effect of magnetic field generated by EC becomes lower leading the induced voltage of the sensing coil to increase.

Figure 13 shows the simulation results of the induced voltage changes of the sensing coils, which is caused by the cracks at the hole edge in the first and second laminate plates when it was propagated from 0 mm to 6 mm along the radial direction of the bolt hole of the unidirectional CFRP. Agreeing with experimental results, the variation in the induced voltage of sensing coil 1 at the top laminate plate raised rapidly when the crack occurred at the top laminate plate, while the induced voltages of the other three sensing coils located at the other laminate plates without cracks remained unchanged. When the crack occurred at the second laminate plate, the variation of induced voltage in sensing coil 2 raised rapidly. Figure 14 shows the simulation results of the induced voltage changes corresponding to the crack with 2 mm in the radial direction propagating in the axial direction along the bolt hole from 0 mm to 20 mm of the unidirectional CFRP. Matching well with the experimental results, the changes in induced voltage increases only when the damage appears in the location of the corresponding sensing coil, whereas the induced voltage of the other sensing coils only changes slightly or unvaried.

## 5. Conclusions

A novel built-in distributed EC sensor network has been developed to monitor the hole-edge damages of composite bolted joints. The built-in distributed EC sensor network contains one exciting coil covering the entire area of the inner wall and four separate sensing coils laying along the axial direction of the bolt hole. Both experiments and numerical simulations were conducted to investigate the performance of the EC array sensing films to quantitatively track the hole-edge damage. Based on the experimental and simulation results, it is obviously that the changes in induced voltage of the sensing coil rises quickly when the hole edge crack propagates in the radial direction of bolt from 0 mm to 3 mm at the laminate plate where the sensing coil is located, whereas the induced voltages of the other sensing coils located at the other laminate plates without cracks only change slightly or remain unchanged. The configuration in which the exciting current in the boundary trace of one sub-coil is in the same direction as that in the adjacent boundary trace of another sub-coil is more beneficial because it can improve the intensity of the induced voltage better than when the adjacent boundary trace currents are in opposite directions. The results demonstrated that the developed EC array sensing film can effectively identify not only whether there is damage at the hole edge, but also the damage location in the thickness and quantitative size. 

The proposed EC array sensing film shows a good promise for real-time monitoring the crack in the composite bolted joint with several benefits, such as lightweight and low effect on original structural integrity. However, the ability of EC array sensing film to quantitatively track the damage in the radial direction needs to be further improved. Additionally, other improvements include how to apply this SHM technology into real engineering applications. 

## Figures and Tables

**Figure 1 materials-12-02785-f001:**
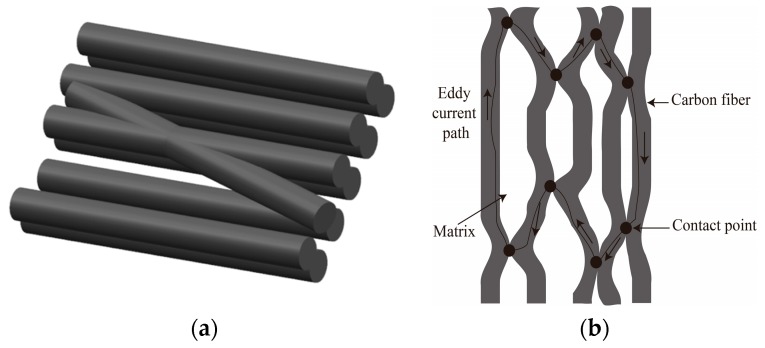
The schematic diagram of conductivity of carbon fiber reinforced plastic (CFRP): (**a**) Contact points between carbon fibers; (**b**) Current path between carbon fibers.

**Figure 2 materials-12-02785-f002:**
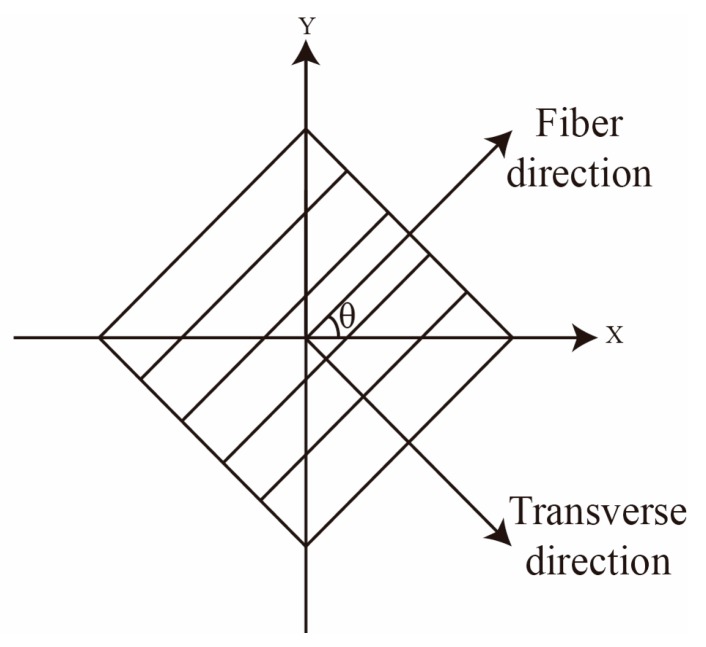
Rotating coordinate system.

**Figure 3 materials-12-02785-f003:**
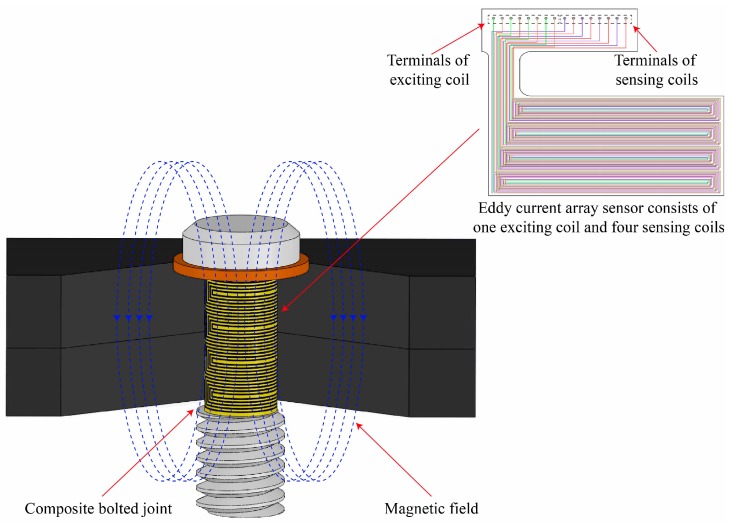
Operating principle of eddy current sensing film mounted on the bolt of the composite joint.

**Figure 4 materials-12-02785-f004:**
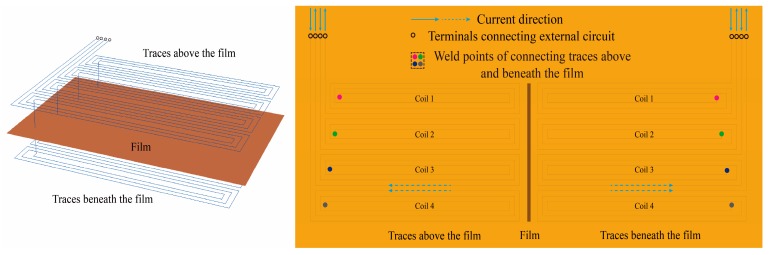
Schematic plot of exciting coil.

**Figure 5 materials-12-02785-f005:**
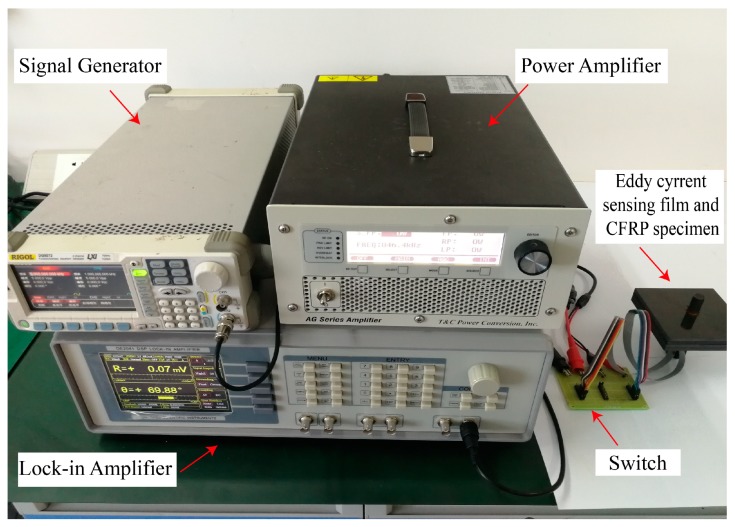
Experimental setup.

**Figure 6 materials-12-02785-f006:**
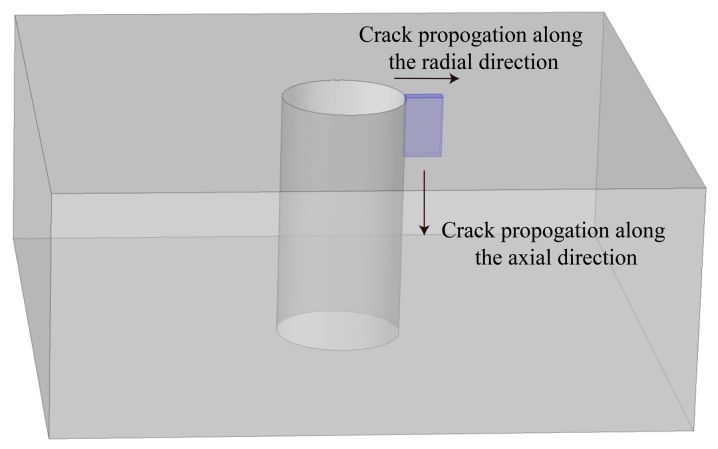
Schematic plot of the artificial crack propagating around radial and axial directions.

**Figure 7 materials-12-02785-f007:**
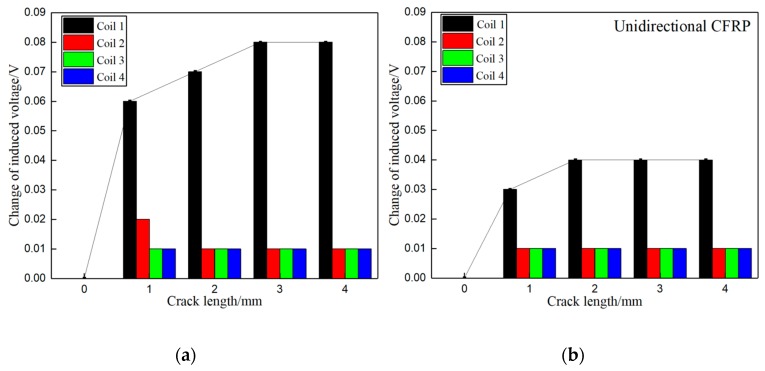
Change of induced voltage caused by the crack in the top laminate plate along the radial direction of bolt hole: (**a**) Exciting coils with the same current direction in boundary traces; (**b**) Exciting coils with the opposite current direction in boundary traces.

**Figure 8 materials-12-02785-f008:**
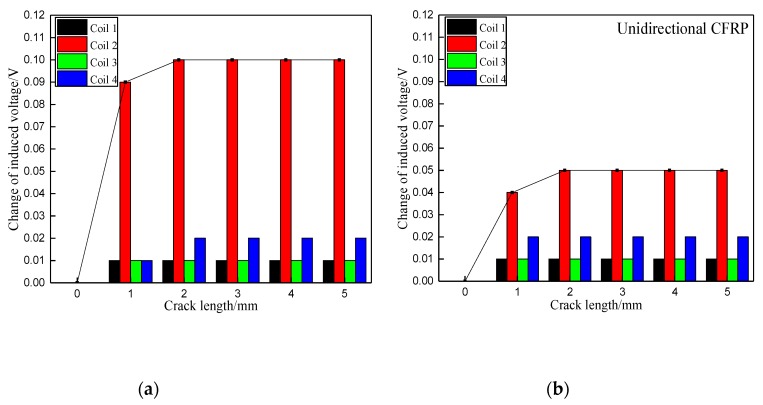
Change of induced voltage caused by the crack in the second laminate plate along the radial direction of bolt hole: (**a**) Exciting coils with the same current direction in boundary traces; (**b**) Exciting coils with the opposite current direction in boundary traces.

**Figure 9 materials-12-02785-f009:**
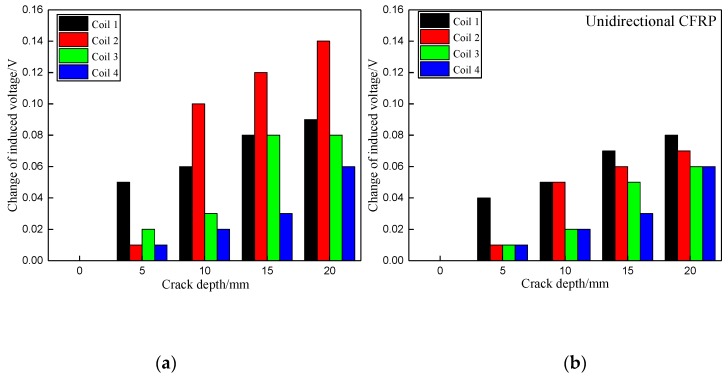
Change of induced voltage of crack growth in the axial direction along the bolt hole: (**a**) With exciting current in the same direction in the boundary between exciting sub-coils; (**b**) With exciting current in the opposite direction in the boundary between exciting sub-coils.

**Figure 10 materials-12-02785-f010:**
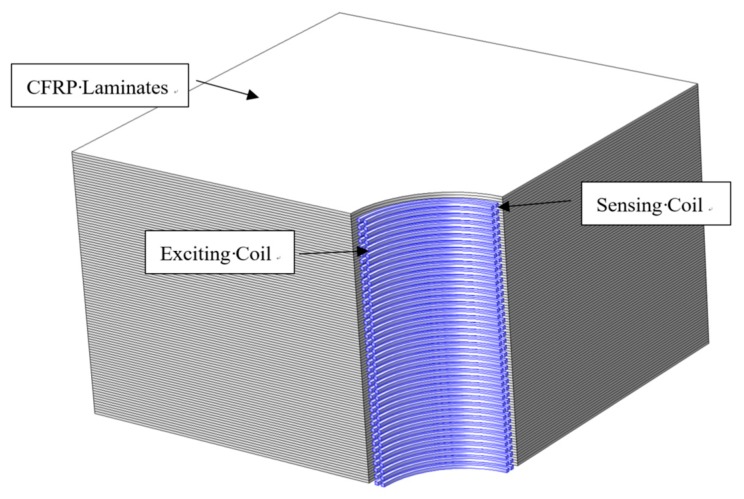
Finite element method (FEM) simulation model.

**Figure 11 materials-12-02785-f011:**
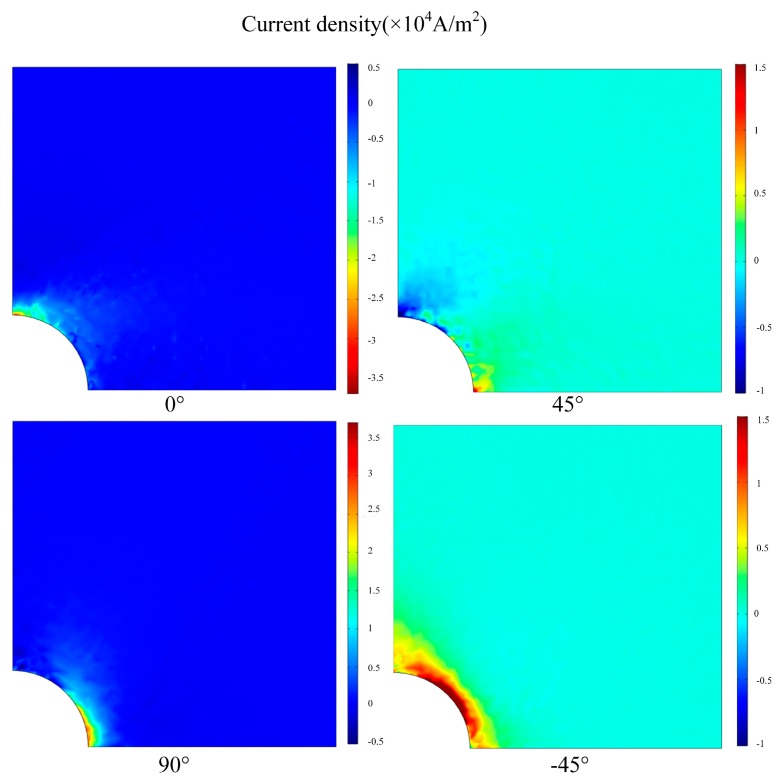
Eddy current distributions on the first four plies [0/45/90/−45] from the top of CFRP composite without damage.

**Figure 12 materials-12-02785-f012:**
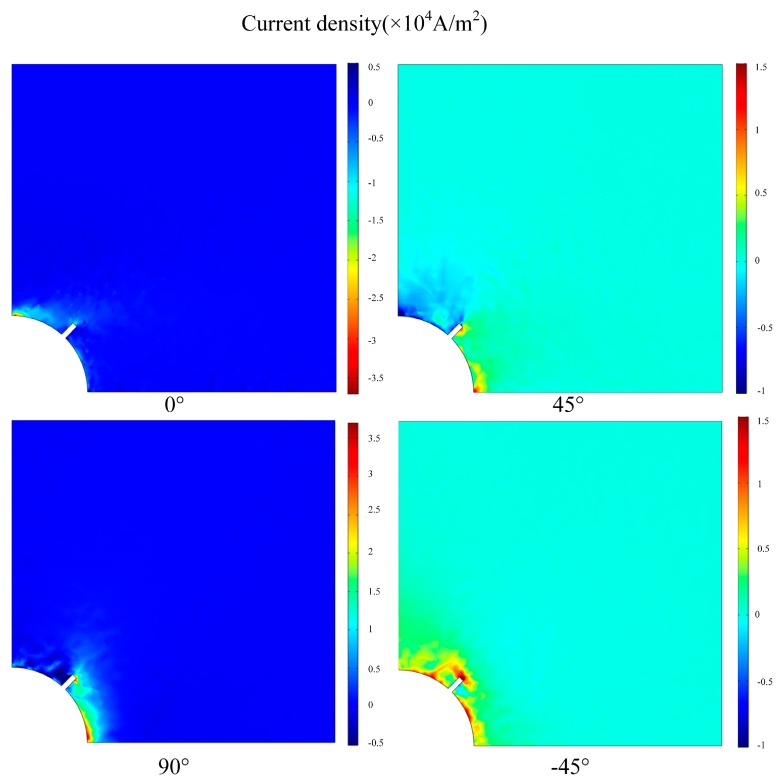
Eddy current distributions on the first four plies [0/45/90/−45] from the top of CFRP composite with an artificial crack.

**Figure 13 materials-12-02785-f013:**
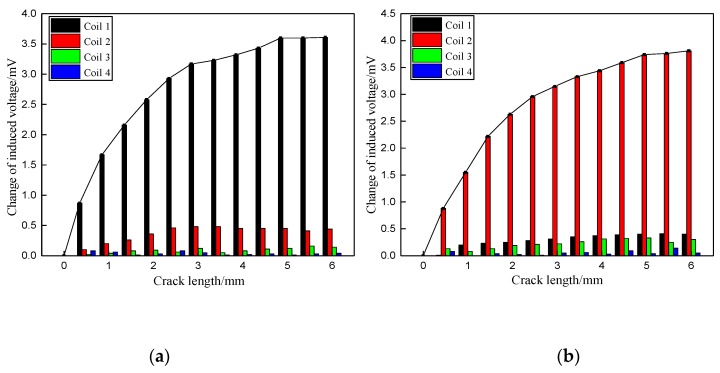
Simulation of induced voltage change caused by the crack in two different laminate plates: (**a**) The top laminate plate; (**b**) The second laminate plate.

**Figure 14 materials-12-02785-f014:**
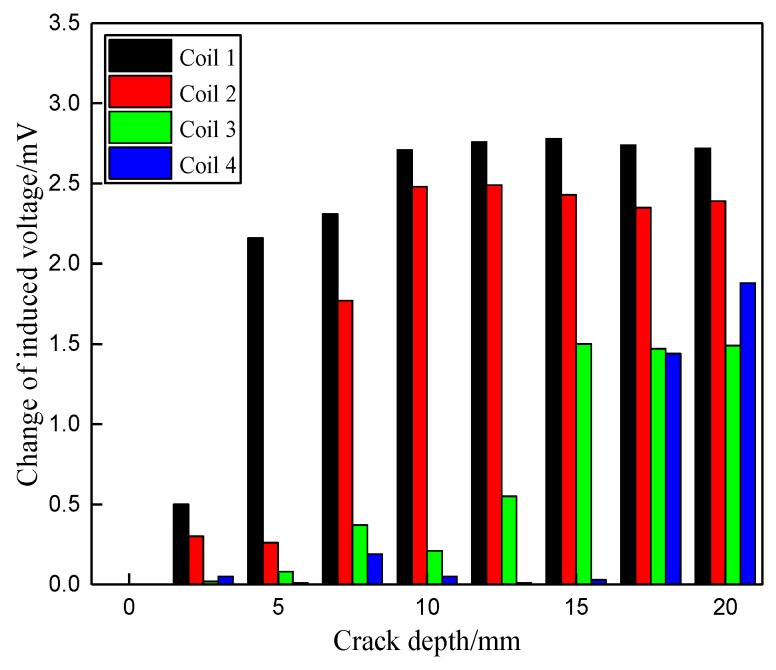
Simulation of the induced voltage change caused by the crack propagating in the axial direction along the bolt hole of the unidirectional CFRP.

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
