# Peer review of "On-Site Health Monitoring of Composite Bolted Joint Using Built-In Distributed Eddy Current Sensor Network"

_materials, 2019, doi:10.3390/ma12172785_

Round 1

Reviewer 1 Report

In lines 110-121 describes the basic principles of monitoring using the eddy current sensor. Exciting coil will excite the alternating voltage in sensing coils also in the first place directly with such a close location of exciting coil and several sensing coils. Was it somehow taken into account in the calculations and was in sight in the experiment? What do blue dashed lines mean in Figure 3? In lines 133-135 need more detail and understandable write about how as all of the exciting traces are in the series connection as one coil. Assuming the most logical option, when the end of one sub-coil is connected to the beginning of the next sub-coil, and for the left and for the right configurations in Figure 4 obtain opposite current direction at the boundary of two adjacent sub-coils. The direction of current at the boundary of two adjacent sub-coils can be obtained the same, only if you apply a different order serial connection sub-coils. Line 149. Can you substantiate why the selected frequency 8 MHz? In lines 173-175 written: “The results indicate that the value of induced voltage is higher when using the exciting sub-coils with the exciting current in the same direction in the nearby traces at the boundary of two adjacent sub-coils comparing to the circumstance of current in the opposite direction.” Given my comment 3, this may look strange. If such different experimental results are obtained for the left and right configurations in Figure 4 with the same method of serial connection sub-coils it is at least suspicious. First, the authors need to be very clear about the question I raised in comment 3.

Reviewer 2 Report

The paper deals with the SHM of carbon fiber bolted joints using eddy current technique.

The work presents both numerical simulations and experimental validation. The results are in decent agreement and the work will be interesting to the readers.

The only comments are:

The schematic of the coils its placement on the structure needs to be improved for clarity. A clearer figure with sections and colors may be useful

Figures 7,8,9 and 13 need more discussion.

For instance the in figure 13 the coil 3 shows reduced voltage when the crack increases from 7.5 to 10 what is the reason for that?

The shortcomings of the method and areas of future work need to be identified.

Also a small discussion on the cost of the sensors and comparison with other techniques may be useful as well.

The paper needs to be thoroughly proof-read as there are significant grammar and typographical mistakes.
